# Fungal–Bacterial Networks in the Habitat of SongRong (*Tricholoma matsutake*) and Driving Factors of Their Distribution Rules

**DOI:** 10.3390/jof8060575

**Published:** 2022-05-27

**Authors:** Rui-Qing Ji, Yang Xu, Yan-Ji Si, Chayanard Phukhamsakda, Yu Li, Li-Peng Meng, Shu-Yan Liu, Meng-Le Xie

**Affiliations:** 1Engineering Research Center of Edible and Medicinal Fungi, Ministry of Education, Jilin Agricultural University, Changchun 130118, China; jiruiqing@jlau.edu.cn (R.-Q.J.); xuyang@shbio.com (Y.X.); siyanjisyj@126.com (Y.-J.S.); chayanard91@gmail.com (C.P.); yuli966@126.com (Y.L.); 2Jilin Forestry Science Institute, Changchun 130032, China; 3Life Sciences College, Northeast Normal University, Changchun 130024, China

**Keywords:** *Tricholoma matsutake*, forestry types, fungi–bacterial networks, mixed forest with *Quercus mongolica*, soil structure, soil physicochemical parameters

## Abstract

Soil origin, mycorrhizal plant partners and environmental factors affect the growth and development of SongRong (*Tricholoma matsutake*). In order to clarify the relationships of fungi–bacteria networks and various influence factors in the habitat of SongRong, we chose three collection sites with a *Quercus mongolica* pure forest (plot A without SongRong was used as the control sample site), *Q. mongolica* mixed *Rhododendron dauricum* (plot B) and *Q. mongolica* mixed with *R. dauricum* and *Pinus densiflora* (plot C). By using high-throughput sequencing, we obtained a total of 4930 fungal and 55501 bacterial amplicon sequence variants (ASVs) based on internally transcribed spacer ribosomal RNA (ITS rRNA) and 16S ribosomal RNA (16S rRNA) sequencing via the Illumina NovaSeq platform. In the habitat soil of SongRong (plot B and plot C), alpha or beta diversity and species compositions of fungi and bacteria were different from plot A. The fungal–bacterial networks follow the selection rule that few dominant genera account for the greater relative abundance. Forest types, but not the host itself, drove the fungal–bacterial networks of the forest soil, and soil physicochemical characteristics and texture affected their abundance. The abundance of *Tricholoma* was affected by the fungal and bacterial abundance in the habitat.

## 1. Introduction

SongRong (*Tricholoma matsutake* Singer) is a well-known ectomycorrhizal fungus (ECMF) that is highly prized for the medicinal values of its edible fruiting bodies [1,2]. It is one of the rarest species in China and is not widely distributed *T. matsutake* is mainly distributed in Heilongjiang Province, Jilin Province, Sichuan Province, Tibet Autonomous Region and Yunnan Province [3,4,5,6]. The mycelial biomass in the below-ground portion of *T. matsutake* plays an important role in the formation of its fruiting body, which is affected by soil origin, host species, climate factors, etc. [7,8]. However, the contributing role of each factor is not fully understood. It is, therefore, necessary to clarify the microbe, host and forest types and environmental factors in the soil habitat of *T. matsutake* and to understand their effects; this will provide the basis for clarifying *T. matsutake*’s growth and development characteristics.

We know that various abiotic and biotic factors are important factors that influence fungi or other microbial structures and their distribution in forest ecosystems [9,10,11,12,13,14,15]. The distribution of *T. matsutake*’s fruiting bodies was closely related to their symbiotic partners, and Pinaceae are believed to be the main host of *T. matsutake*. In addition, there are a few reports about the fruiting bodies of *T. matsutake* found in the Fagaceae forest [1,8,16,17,18]. Matsushita et al. studied the molecular sequence of *T. matsutake* from *P. densiflora* forests in Japan, *Castanopsis orthacantha* forests and *Quercus* spp. forests, *Pinus yunnanensis* forests in China, and *T. nauseosum* from *P. densiflora* forests in Europe, and the results showed that not only *T. matsutake* of China from a broad-leaved tree was identical to that of Japanese *T. matsutake* but also that the differences of *T. nauseosum* in Europe were generally considered to be the same species [19]. It was reported that *T. matsutake* usually grows in Japanese red pine (*Pinus densiflora*) forests, where it forms a symbiotic relationship via underground hyphae with the roots of Japanese red pine trees [20,21,22,23]. In China, the primary hosts of *T. matsutake* are *P. densiflora*, *Quercus mongolica* and *Q. aquifolioides*, and its fruiting bodies are often found in mixed forests of these species [8,24]. In South Korea, most *T. matsutake* was found in the forests of *P. densiflora* mixed *Quercus* species and Rhododendron species [25]. Little is known about the influence of forest types on underground microorganisms of the *T. matsutake* habitat compared with their hosts.

Soil can be seen as a factor that directly affects microbial communities and the structures within them [26,27]. *T. matsutake* in Japan appears in nutrient-poor soils, such as granite, chert and sandstone, which are acidic, with pH values of 4.5–5.0 [28]. In northeast China, *T. matsutake* grows in soil with a humic layer, as well as dry and loose acid soil [29]. The soil pH is the primary factor affecting the evenness of underground mycorrhizal communities [30]. Additionally, the complementarity of phosphorus uptake is affected by the diversity of ectomycorrhizal fungal species, and the efficiency of the phosphorus uptake decreases when the soil moisture is limited [31]. Soil texture is the main driver of biodiversity and acts on microbial and macroscale fauna communities through different mechanisms [32,33,34,35]. The soil microbial community is a key member of maintaining soil microecological balance, the correlation of which constrains the change in abundance of each member [36,37]. Despite the complexity of the food web of soil microorganisms, some correlation results have been reported as technology has improved. For example, analyses of the linear regression equation showed that the relative abundance of *Russula* was correlated with *Lachnum* (most species of which are saprophytic fungi) and *Ilyonectria* (some species of which can cause root rot disease) in a pure *Q. mongolica* forest [11]. In addition, the relative abundance of *Suillus* was shown to have certain relations with *Umbelopsis* in a Korean pine forest [12]. Some studies reported on the composition of soil microorganisms in the *T. matsutake* habitat [38,39,40], but there are conflicting accounts of the specific microbial groups in the different habitats of *T. matsutake*. Therefore, we chose two habitats of *T. matsutake* as sampling sites, which are covered by *Q. mongolica* mixed with other trees or shrubs, and one control site without fruiting bodies of *T. matsutake* covered by pure *Q. mongolica* at the same latitude.

This study compares the distribution of the fungi and bacteria communities in the habitat of *T. matsutake* under different forest types in northeast China; compares the soil texture, soil physicochemical characteristics and the effects on the fungal–bacterial networks of three collection sites; and quantitatively assesses the relative importance of correlations of fungal and bacterial species in the soil with respect to the distribution of *Tricholoma*. We analyzed the samples at the genera level due to the limitations of species identified by second-generation amplicon sequencing. We also analyzed the composition of the species of *Tricholoma*, but the results were not confirmed due to limitations with the sequencing technology.

## 2. Materials and Methods

### 2.1. Study Site Selection

In this study, we selected three sampling sites in northeast China. The control site, named plot A, is covered by a pure forest of *Quercus mongolica* (PA, 132°3′2″ E, 46°7′20″ N, asl. 230 m) (Appendix A). Plot B, located in Baoqing County, Heilongjiang Province (PB, 132°3′3″ E, 46°7′16″ N, asl. 280 m), consists of *Q. mongolica* mixed with *Rhododendron dauricum* and has no other woody plants (Appendix A). Plot C (PC, 129°39′52″ E, 42°33′31″ N, asl. 380 m), located in Longjing County, Jilin Province, is a National Nature Reserve of SongRong (*T. matsutake*), consists of a mixture of *Q. mongolica*, *R. dauricum* and *Pinus densiflora* and has no other woody plants (Appendix A).

### 2.2. Sampling Strategy and Processing

We marked five quadrants measuring 20 m × 20 m within a space of more than 100 m. The specimens were sampled by a five-point sampling method [11] in June–July and August–September (the fruiting season of *T. matsutake*). All soil subsamples from the same plot and same sampling time were homogenized and pooled to obtain a composite sample, which was placed into a cooler containing dry ice and transported to the laboratory. Then, it was placed in a −70 °C refrigerator for DNA extraction and high-throughput sequencing.

### 2.3. DNA Extraction and Illumina NovaSeq Platform

Total microbial genomic DNA samples were extracted using the DNeasy Power Soil Kit (QIAGEN, Inc., Hilden, Germany) following the manufacturer’s instructions. The DNA samples were stored at −20 °C until further analyses. The quantity and quality of extracted DNA samples were measured using a NanoDrop ND-1000 spectrophotometer (Thermo Fisher Scientific, Waltham, MA, USA) and agarose gel electrophoresis, respectively.

PCR amplification of the bacterial 16S rRNA gene V3-V4 region was performed using primers 338F and 806R [41]. PCR amplification of the fungal ITS1 region was performed using primers ITS5 and ITS2 [42]. PCR amplicons were purified with AgencourtAMPure Beads (Beckman Coulter, Indianapolis, IN) and quantified using the PicoGreen dsDNA Assay Kit (Invitrogen, Carlsbad, CA, USA). After the individual quantification step, amplicons were pooled in equal amounts, and paired-end 2 × 300 bp sequencing was performed using the Illumina NovaSeq platform at Shanghai Personal Biotechnology Co., Ltd. (Shanghai, China).

### 2.4. Analysis of Physical/Chemical Parameters of Soil Samples

In order to measure soil parameters, soil samples that were completely dried with silica beads for about 1–2 weeks were screened with a 2 mm sieve to remove stone particles. The soil available phosphorus (SAP) was determined using the 0.5 mol·L^−1^ NaHCO3 extraction-molybdenum anti-colorimetric method. The soil organic matter (SOM) was detected using the potassium dichromate volumetric method. The soil effective nitrogen (SEN) was measured using the alkali-diffusion method. The soil available potassium (SAK) was determined using 1 mol·L^−1^ NH4OAC extraction-flame photometry [43]. The soil pH (SpH) was measured by potentiometry.

We determined the soil texture according to the international soil texture classification standards [44]. First, we removed SOM and carbonate by adding hydrogen peroxide (10%, GR) and hydrochloric acid (0.2%, GR) to 250 mL beakers of 10 g sieved soil (2 mm sieve). Next, 10 mL of sodium hydroxide solution (0.05 mol·L^−1^, GR) was added as a dispersant, and the beakers were shaken evenly before analysis. Finally, the soil sample was washed with water in a sedimentation cylinder. The soil composition was measured using the pipette method. The soil texture type was determined using the contents of loamy clay (particle size < 0.002 mm), silty clay (0.02–0.0002 mm) and sandy clay (0.02–2 mm).

### 2.5. Bioinformatics

Microbiome bioinformatic analyses were performed using QIIME2 2019.4 [45] with slight modifications according to the reference. Raw sequence data were demultiplexed using the demux plugin and then primer-cut with the cutadapt plugin [46]. Next, the sequences were filtered for quality, denoised, merged and removed from chimaera using the DADA2 plugin [47]. Non-singleton amplicon sequence variants (ASVs) were aligned with MAFFT [48] and used to construct a phylogeny with FastTree2 [49]. Taxonomy was assigned to ASVs using the Naïve Bayes classification technique in the feature-classifier plugin [50] against the SILVA Release 132/UNITE Release 8.0 Database [51].

### 2.6. Data Processing and Statistical Analysis

A one-way analysis of variance (ANOVA) was performed to analyze differences using SPSS 19.0 software in the phylum classification order richness. All statistical analyses were assessed at α = 0.05 or α = 0.01.

Spearman correlation analysis, RDA or CCA was used to analyze the relationships between soil factors and microbial diversity and composition [52]. A one-way ANOVA and a Tukey’s HSD test at a 5% level of significance was used to test the effect of forest development on soil properties and microbial characteristics. RDA, CCA and ANOVA were conducted using the program R 3.1.1.

### 2.7. Fungal and Bacterial Communities

β-diversity distances were used to describe the differences between samples. We used NMDS analysis, which ranks fungal community sample distances by calculating the Jaccard difference matrix at the level of ASVs. NMDS analysis is not affected by the numerical value of sample distance and only considers the size relationships between samples. When the stress value (Stress) is less than 0.2, the result of NMDS analysis is more reliable [53].

We used the Chao1 index [54] and the Simpson index [55] to assess the fungal and bacterial alpha diversity. The higher the Simpson index value, the higher the community diversity. We used the ASVs abundance table to create Venn diagrams that describe the differences in ASVs among the samples. Genera with a relative abundance of ASV ≥ 1.00% were considered dominant [56]. We used LDA Effect Size (LEfSe) analysis to assess the different robust species between groups, namely biomarkers, which combined the non-parametric Kruskal–Wallis and Wilcoxon rank-sum tests with LEfSe [57].

We used PICRUSt2 (Phylogenetic Investigation of Communities by Reconstruction of Unobserved States) [58] to predict the functions of the fungal or bacterial community in the habitat soil of *T. matsutake*. PICRUSt2 is the retrieval and prediction of ITS genes and 16S gene sequences in the MetaCyc (https://metacyc.org/, accessed on 30 August 2020) functional database. MetaCyc has over 1400 pathways, including metabolic pathways and enzymes from all domains of life. The pathways in MetaCyc are curated from the primary scientific literature and are experimentally determined small-molecule metabolic pathways. Because MetaCyc contains only experimentally elucidated knowledge, it provides a uniquely high-quality resource for metabolic pathways and enzymes. The abundance values of primary and secondary metabolic pathways were obtained from functional units from soil fungi or bacteria according to the above-mentioned database of metabolic pathways and certain calculation methods. Heat maps and bar charts were then made according to the abundance values of metabolic pathways to present the abundance of secondary pathways in the samples, which were then clustered.

## 3. Results

### 3.1. Bacterial Community in the Soil of T. matsutake Habitat

We obtained bacterial 16S rRNA gene amplicons 98,157, 117,064 and 105,312 from PA, PB and PC, respectively, which were divided into 20,661, 19,464 and 25,549 ASVs with 100% similarity. NMDS analysis ranked the bacterial community sample distances by calculating the Jaccard difference matrix at the ASV level. The stress value was 0.0317 (<0.2, Figure 1a), so the result is reliable. The bacterial communities from the three plots were separated, and the distance between the PA and PC was relatively close.

The highest Chao1 index of soil bacteria was shown in PC, which was significantly different from PB (*p* < 0.05) and was not significantly different from PA (*p* > 0.05). These results are similar to the NMDS. The Simpson index of PA was the highest, but with no significant difference (*p* > 0.05; Figure 1b). There were 2017 ASVs shared among the three collection sites. There were 18,113 (70.9%) unique ASVs in PC and 13,757 (70.7%) ASVs in PB (Figure 1c).

Based on the Silva database, all bacterial ASVs were clustered and classified into 41 phyla, 119 classes, 284 orders, 481 families and 1021 genera. The cluster diagram of the top 20 genera in terms of relative abundance shows that the species of PB and PC were closer but different from the abundance of PA. There were 10 dominant genera in plot A (relative abundance ≥ 1%), which accounts for 21.7% of the relative abundance of known genera. There were 14 dominant genera in plot B, which accounts for 40.2% of the relative abundance of known genera, and there were 16 dominant genera in plot C, which account for 34.6% of the relative abundance of known genera (Appendix A). The first 20 genera from the three plots were compared (Figure 1d). The genera with the highest relative abundance (>1%) in plot A were *KD4-96*, *Candidatus_Udaeobacter*, *Subgroup_6*, *Allorhizobium-Neorhizobium-Pararhizobium-Rhizobium*, *Bradyrhizobium* and *Thiobacillus*. The genera *Burkholderia-Caballeronia-Paraburkholderia*, *Subgroup_2*, *Acidothermus*, *Bryobacter*, *Candidatus_Solibacter*, *Roseiarcus, AD3*, *Granulicella*, *Candidatus_Xiphinematobacter* and *Conexibacter* had the highest relative abundance (>1%) in plot B. *Bradyrhizobium*, *Mycobacterium*, *RB41*, *Sphingomnas* and *IMCC26256* had the highest relative abundance (>1%) in plot C.

Fifty bacterial taxa were differentially abundant among the three sample groups. The biomarkers of plot A were *Thiobacillus* (2.86%), *Sulfurovum* (2.02%) and *Vibrionimonas* (1.70%); the biomarkers of plot B were *Burkholderia-Caballeronia-Paraburkholderia* (12.62%), *Acidothermus* (2.50%), *Subgroup_2* (5.25%), *Acidothermus* (2.50%), *Bryobacter* (2.48%), *Candidatus_Solibacter* (1.78%), *Roseiarcus* (1.70%), *Granulicella* (1.48%) and *WD260* (1.45%); and the biomarker of plot C was *Mycobacterium* (2.54%) (Appendix A).

### 3.2. Fungal Community in the Soil of T. matsutake Habitat

We obtained 91,057, 88,663 and 78,029 effective fungal ITS amplicons from PA, PB and PC, respectively. The corresponding amplicons were divided into 2493, 1523 and 2101 ASVs with 100% similarity.

NMDS was used to analyze the fungal community sample distances by calculating the Jaccard difference matrix at the ASVs level. The stress value was 0.0167 (<0.2, Figure 2a). The fungal communities in the soil of the *T. matsutake* habitat (PB and PC) and control site (PA) were separated, and the distance between PB and PC was relatively close.

Alpha diversity reveals the highest Chao1 index in PA but no significant difference from those of PB and PC (*p* > 0.05). The Simpson index of these three plots also showed the same results (*p* > 0.05, Figure 2b). Only 211 ASVs were shared among the three collection sites. The highest number of unique fungal ASVs were detected in PA (1 945 ASVs, 78.2%) and the lowest number in PB (930, 61.1%). PB and PC shared 413 ASVs, which is the highest number of unique fungal ASVs (Figure 2c).

Based on the UNITE database, all the fungal ASVs were clustered and classified into 11 phyla, 44 classes, 105 orders, 222 families and 423 genera. Comparatively, the species compositions of PB and PC were more similar. Among them, four ectomycorrhizal fungi (ECMF) genera (*Cortinarius*, *Sebacina*, *Tomentella* and *Russula*) and one saprotrophic fungi (SAF) genus (*Mortierella*) had the highest relative abundance in PA. Five ECMF genera (*Tricholoma*, *Hygrophorus*, *Sistotrema*, *Lactarius* and *Cenococcum*) and three SAF genera (*Trichoderma*, *Umbelopsis* and *Penicillium*) had the highest relative abundance in PB. Two ECMF genera (*Lactifluus* and *Hydnellum*), one ericoid mycorrhizal fungi (ERMF) genera (*Oidiodendron*), one SAF genus (*Cladophialophlora*) and three other fungal genera (*Delastria*, *Geminibasidium*, and *Saitozyma*) had the highest relative abundance in PC (Figure 2d,e). There were 5, 11 and 14 dominant genera (the relative abundance ≥ 1%) in PA, PB and PC, which account for 83.6%, 83.7% and 78.8% of the relative abundance of known genera, respectively. *Tricholoma* was the dominant genus in PB (21.84%) and PC (1.87%) (Appendix A).

The top fifty fungal taxa of the scores of LDA were shown to be differentially abundant among the three sample groups. The fungal biomarker genera were *Sebacina* (4.63%), *Inocybe* (0.73%) and *Capronia* (0.29%). The biomarker genera in plot B were *Tricholoma* (23.55%), *Umbelopsis* (13.5%), *Penicillium* (2.0%), *Piloderma* (0.56%), *Vanrija* (0.32%) and *Cutaneotrichosporon* (0.24%). The biomarker genera of plot C were *Hydnellum* (18.78%), *Lactifluus* (2.46%), *Cladophialophora* (1.12%), *Oidiodendron* (0.93%), *Tausonia* (0.67%), *Solicoccozyma* (0.64%), *Amphinema* (0.55%), *Tomentellopsis* (0.35%) and *Phialocephala* (0.23%) (Appendix A).

### 3.3. Soil Physicochemical Conditions and Their Effects on the Distributions of the Fungal and Bacterial Community

The contents of phosphorus content (SAP) and soil organic matter (SOM) in PA were significantly (*p* < 0.05) higher than those of PB and PC. Other parameters, e.g., soil pH (SpH) and nitrogen content (SEN), were shown to be significantly (*p* < 0.05) different among three sites, and the content of potassium content (SAK) showed no significant differences among the three sampling sites (Table 1). The relationship between community compositions and environmental factors was analyzed using RDA. For bacteria, the RDA axes 1 and 2 account for 54.63% and 11.58% of the total variation (Figure 3a). The soil pH (*p* = 0.001), soil organic matter (*p* = 0.044) and nitrogen content (*p* = 0.018) had significant difference (*p* < 0.05) correlations with the bacterial abundance. For fungi, the RDA axes 1 and 2 accounted 41.07% and 18.42% of total variation (Figure 3b). The soil pH (*p* = 0.001), soil organic matter (*p* = 0.002), potassium content (*p* = 0.001), nitrogen content (*p* = 0.001) and phosphorus content (*p* = 0.005) had (*p* < 0.01) correlations with the fungi abundance with highly significant differences.

The contents of clay and silt were higher in PA, but PC and PB had higher sand contents. The results showed that the soil textures of PA, PB and PC were loamy clay, sandy clay and sandy clay, respectively (Figure 4). The relationships between community compositions and soil texture were analyzed using CCA. For bacteria, the CCA axes 1 and 2 accounted for 24.23% and 4.22% of the total variation, respectively (Figure 5a). The clay content (*p* = 0.007), silt content (*p* = 0.003) and sand content (*p* = 0.015) had highly significantly different correlations (*p* < 0.01) with bacterial abundance. For fungi, the CCA axes 1 and 2 accounted for 23.32% and 17.31% of the total variation, respectively (Figure 5b). The clay content (*p* = 0.001), silt content (*p* = 0.001) and sand content (*p* = 0.001) had highly significantly different correlations (*p* < 0.01) with fungal abundance.

### 3.4. The Community of Tricholoma in Three Collection Sites and the Influence Factors

The relative abundance of *Tricholoma* was highest in PB with 21.84%, PC with 1.87% and PA with 0.01% (Appendix A). Among them, the relative abundance of *T. matsutake* in *Tricholoma* was highest in PB with 97.06% (Appendix A).

The RDA results showed that soil pH, soil organic matter, nitrogen content and phosphorus contents were negatively correlated with the relative abundance of *Tricholoma*, while potassium content was positively correlated (Figure 3b). Sand and clay contents showed a positive correlation with the relative abundance of *Tricholoma*, while silt content showed a negative correlation (Figure 5b).

Soil microbes were shown to have certain relations with *Tricholoma* by Spearman analysis. The fungal genera, *Umbelopsis* (r = 0.946, *p* = 0.000), *Trichoderma* (r = 0.809, *p* = 0.000), *Leptodontidium* (r = 0.706, *p* = 0.001), *Penicillium* (r = 0.919, *p* = 0.000), *Mucor* (r = 0.73, *p* = 0.001), *Dasyspora* (r = 0.649, *p* = 0.004) and so on (19 genera are basically saprophytes) were positively correlated with *Tricholoma*. Nine fungal genera, namely *Russula* (r = −0.832, *p* = 0.001), *Inocybe* (r = −0.710, *p* = 0.001), *Leotia* (r = −0.781, *p* = 0.000), *Entoloma* (r = −0.666, *p* = 0.003), *Trechispora* (r = −0.695, *p* = 0.001), *Cylindrosympodium* (r = −0.638, *p* = 0.004), *Ilyonectria* (r = −0.695, *p* = 0.001) and *Clitocybe* (r = −0.617, *p* = 0.006) were negatively correlated with *Tricholoma* (Appendix A). Most of them are ectomycorrhizal fungi or endophyte.

Furthermore, there are 142 bacterial genera in the habitat soil, such as *Burkholderia-Caballeronia-Paraburkholderia* (r = 0.866, *p* = 0.000), *Bradyrhizobium* (r = 0.676, *p* = 0.002) and *Mycobacterium* (r = 0.603, *p* = 0.008), that were positively correlated with *Tricholoma*. Other bacterial genera, such as *MND1* (r = −0.633, *p* = 0.005), *Phyllobacterium* (r = −0.835, *p* = 0.000), *Pseudoxanthomonas* (r = −0.791, *p* = 0.000), *Nitrospira* (r = −0.749, *p* = 0.000), *Agromyces* (r = −0.611, *p* = 0.007), *Thermomonas* (r = −0.717, *p* = 0.001), *Bosea* (r = −0.632, *p* = 0.005), *Luteimonas* (r = −0.655, *p* = 0.003), *Sphingobium* (r = −0.638, *p* = 0.004), *Taibaiella* (r = −0.618, *p* = 0.006) and *Streptococcus* (r = −0.761, *p* = 0.000) had significant (*p* < 0.01) negative correlations with *Tricholoma* (Appendix A).

### 3.5. Functional Potential Analysis of Soil Fungi and Bacteria in the Habitat of T. matsutake

Based on MetaCyc metabolic database, we used PICRUSt2 software to predict and analyze the metabolic function of soil fungal communities and obtain the abundance of metabolic pathways. In the three study sites, soil bacteria had 7 metabolic pathways at the first level and 61 secondary metabolic pathways: a biosynthesis pathway (12 secondary metabolic pathways), degradation/use/assimilation and energy metabolic pathways (16 secondary metabolic pathways), precursor metabolites of the generation of metabolites and energy (17 secondary metabolic pathways), glycan pathways (2 secondary metabolic pathways), metabolic clusters (10 secondary metabolic pathways), detoxification (2 secondary metabolic pathways), and macromolecule modification (2 secondary metabolic pathways) (Appendix A). Soil fungi have 5 metabolic pathways at the first level and 29 secondary metabolic pathways: biosynthesis pathway (7 secondary metabolic pathways), degradation/use/assimilation and energy metabolic pathways (7 secondary metabolic pathways), generation precursor metabolites of the generation of metabolite and energy (9 secondary metabolic pathways), glycan pathways (1 secondary metabolic pathways) and metabolic clusters (5 secondary metabolic pathways) (Appendix A).

In order to understand which species encode genes with certain functional potential, the composition of species involved in tertiary metabolic pathways was analyzed using stratified sample metabolic pathway abundance tables, with the assumption that all or most of these metabolic pathways can be independently performed by the species. When PA was taken as the control, 13 bacterial metabolic pathways in PB were differently (*p* < 0.05) up-regulated (Appendix A). The functions of the dominant genus of bacteria or fungi were further analyzed. *Burkholderia-Caballeronia-Paraburkholderia* took part in 11 up-regulated metabolic pathways: GALACTARDEG-PWY, GLUCARGALACTSUPER-PWY, GLUCARDEG-PWY, CATECHOL-ORTHO-CLEAVAGE-PWY, PWY-5178, PWY-5417, PWY-5430, PWY-5431, PWY-6182, PWY-7295 and PWY-722. *Candidatus_Solibacter,* the other dominant genera in PB, took part in GALACTARDEG-PWY, GLUCARDEG-PWY and GLUCARGALACTSUPER-PWY. When PA was taken as the control, two bacterial metabolic pathways in PC were differently (*p* < 0.05) up-regulated (Appendix A). Including PWY-722, which was differently up-regulated in the PB/PA group; the other was PWY-6993, which is the pyrrolidine pathway for the degradation of nicotine. Comparative analysis of fungal metabolic pathways in PB/PA or PC/PA showed that only the genera *Penicillium* took part in the only differently (*p* < 0.05) up-regulated metabolic pathways, which was LIPASYN-PWY (Appendix A).

## 4. Discussion

In northeast China, we found that fruiting bodies of *T. matsutake* are primarily distributed in *Q. mongolica* dominant forests mixed with *R. dauricum* or *P. densiflora* and *R. dauricum*, but not in the forests with only *Q. mongolica* trees. In this study, we analyzed bacterial and fungal community composition in the soils from two *T. masutake* habitats (plot B, PB; and plot C, PC) and one control site (plot A, PA) using a high-throughput sequencing approach. The bacterial community of PC was shown to be significantly higher (*p* < 0.05) than PB in terms of the α-diversity indexes; meanwhile, the bacterial composition of PC was close to PA compared with PB according to β-diversity analysis. The fungal community was separated by β-diversity analysis, showing that the fungal species composition in PB clustered with PC and was different from PA. Therefore, soil fungi–bacterial networks in the habitat soil of *T. matsutake* (PB and PC), including species abundance, diversity or species compositions, were different from the control site (PA), in addition to different vegetation.

Host plant species were reported to affect root or rhizospheric soil microbiome, including some ectomycorrhizal fungi (ECMF) [59,60]. *T. matsutake* is a kind of ECMF with relatively specific hosts. Most reports identify Pinaceae as its host [61]; however, *T. Matsutake* has indeed been reported in a few cases in mixed oak forests [1,8,16,17,18,19]. In our research, the fruiting bodies from PB (ON442550) and PC (ON53676) were all identified as *T. matsutake* species according to ITS sequencing. However, we did not obtain the fruiting bodies from PA. Moreover, according to years of experience from local collectors, *T. matsutake* is not found in PA but is found in PB covered by *Q. mongolica*-mixed *R. dauricum* without other woody plants. Only if the ECM roots of *T. matsutake* are found on the roots of deciduous trees in the natural/seminatural habitat, and if the taxonomic identity of both is known, we can confirm that this deciduous tree is the host of *T. matsutake.* Therefore, Further laboratory data are needed to determine whether *Q. mongolica* can be used as a host of *T. matsutake*. However, our results show that PB and PC covered by *Q. mongolica* and mixed with other trees form the habitat of *T. matsutake*; meanwhile, the abundance of *Tricholoma* in PB and PC was shown to be higher than in PA covered by pure *Q. mongolica* forest (Figure 2d). Therefore, different fungi–bacterial communities in the habitat of *T. matsutake* were presumed to be determined by forest type; these results are in agreement with other studies showing that forest type and soil properties drive soil fungal composition [62]. Compared with the effects of a pure forest, the complex forest type causes high diversity and abundance of microbes in the soil [63,64]. In the context of *Q. mongolica* as a host for *T. matsutake*, the forest type in which it is mixed, for example, with *R. dauricum* rather than a pure forest, is more conducive to *T. matsutake* or *Tricholoma*. *R. dauricum* may affect the soil and then, indirectly, the environmental conditions for ECM fungi (including *T. matsutake*). Otherwise, suitable habitats for *T. matsutake* can overlap with suitable habitats for *R. dauricum*. It was reported that *T. matsutake* is usually found in *Rhododendron* spp. [1,16].

The structure of the soil microbial community is determined by numerous biotic and abiotic factors such as plant community composition and soil chemistry (e.g., soil organic C, nutrients, pH) [65,66,67,68]. The distributions of fungi (the relative abundance of dominant fungal genera >78%) and bacteria (the relative abundance of dominant bacterial genera >21%) in three collection sites were not uniform, which revealed that a few dominant species are responsible for the higher relative abundance (Appendix A). In the Carpinus cordata forest, *Inocybe*, *Sebacina*, *Tomentella* and *Russula* account for 29% of the relative abundance [69]. The community compositions of EMF showed the same pattern that a few dominant genera account for the majority of the relative abundance in different forests [70]. This distribution pattern of fungal or bacterial communities is found in other environmental conditions [71].

Our results show that the soil pH, soil organic matter, nitrogen content, potassium content and phosphorus content are similar in PB to that in PC. These characteristics have significant correlations with the abundance of the fungal community. The soil pH, soil organic matter and nitrogen content have significant correlations with the abundance of the bacterial community. Generally, soil pH and nitrogen content are thought to drive the soil fungal–bacterial networks on a global scale or in a specific region [72,73]. Moreover, nitrogen is a key element controlling the species composition, diversity and productivity of many terrestrial ecosystems [68]. It was reported that nitrogen deposition is negatively correlated with some fungal and bacterial communities in the soil [74]. Compared with PA, the habitats PB and PC, which are the habitats of *T. matsutake* and have a higher abundance of *Tricholoma,* showed low pH, soil organic matter, nitrogen content and potassium content, which is consistent with previously reported results about the habitat of *T. matsutake* [8,24,29].

The relationship between microbes in the soil is extremely complex and shows that different microbes restrict or promote each other [75,76]. In PB, *Tricholoma* is the biomarker genera, and the bacterial diversity is lower (*p* < 0.05) than that of PC and PA, but the fungal diversity was shown to be similar to that of PC and PA. In addition, the fungal and bacterial compositions are different from the others (Figure 1 and Figure 2). This is consistent, to some extent, with previous research showing that the active mycorrhizal zone of the shiro of *T. matsutake* has antimicrobial activities, as the abundance of bacteria and sporulating fungi is decreased within that zone [77]. We further identified some fungi or bacteria in the habitat soil of *T. matsutake* that has certain correlations with the *Tricholoma* (Appendix A). The ones that are positively correlated with *Tricholoma* are small fungi with saprotrophic features, such as *Umbelopsis* or *Trichoderma,* which were reported *to be* the dominant genera of some mycorrhizal fungi. *Trichoderma* can increase the success rate of ectomycorrhizal infection [78]. *Umbelopsis*, which has the second-highest relative abundance biomarkers in PB, was also the dominant genus in the habitat of *Tricholoma* in Gyeongju in Southern Korea [79]. The others that are negatively correlated with *Tricholoma* include some large fungi with symbiotic characteristics, such as *Russula*, *Clitocybe* or *Leotia*, some species of which are ectomycorrhizal fungi. The interesting question is, therefore, whether most mycorrhizal fungi compete with each other to some degree while growing cooperatively with some smaller fungi. More studies in this area are needed to confirm this hypothesis. Fortunately, such reports are on the rise. Zhou et al. reported that the domination of *T. matsutake* in shiro causes a lower fungal diversity and specific fungal genera shaped by *T. matsutake* [80].

As shown by the results of our potential functional analysis, only the LIPASYN-PWY was shown to be significantly up-regulated in PB/PA or PC/PA, and it was noted that only *Penicillium* is involved in this pathway, which was the dominant genera in PB. Phospholipases are a class of enzymes present in *P. notetum* and other living beings that hydrolyze glycerophospholipids [81]. *Burkholderia*, *Bradyrhizobium* and *Pseudomonas* were verified by mycorrhizal helper bacteria and had a nitrogen-fixing effect [82,83]. For other metabolic pathways analyses, *Burkholderia* is characterized by the specificity of degradation pathways of carbohydrates, aromatic substances and phenols; it was reported that *Burkholderia* sensu lato (s.l.) species have a versatile metabolism [84]. Among them, PWY-7259 is the L-arabinose (the main component of the hemicellulose plant cell wall material) degradation pathway, which was found in many symbiotic bacteria [85]. The D-glucarate pathway, GALACTARDEG-PWY, GLUCARGALACTSUPER-PWY, GLUCARDEG-PWY and CATECHOL-ORTHO-CLEAVAGE-PWY are correlated with the L-arabinose metabolism pathways [86,87]. The degradation of the plant cell wall is a prerequisite for the establishment of a mycorrhizal relationship, so *Burkholderia* may assist with the mycorrhizal formation in this way, although further tests are needed to verify them. In this paper, some bacteria genera were predicted that have certain positive effects on the abundance of *Tricholoma*. However, the specific functions need further experimental verification.

## 5. Conclusions

In this study, high-throughput sequencing ITS1 for fungi and 16S for bacteria were used to reveal the soil fungi–bacterial networks associated with *T. matsutake*. We found that the above-ground distribution of fruiting bodies reflects the biomass of *T. matsutake* below the ground. Moreover, the distribution of fungal and bacterial communities follows the selection rule that few dominant genera account for the greater relative abundance. The fungal communities in the soil of *T. matsutake*’s habitat are different from the control site in terms of β-diversity and species composition. The bacterial community in the soil with a higher abundance of *Tricholoma* shows a difference from that with a lower abundance of *Tricholoma*. Compared with the effects of the host *Q. mongolica* itself, the forest type of *Q. mongolica* drives the soil microbial composition. Generally, the content of available nitrogen, available phosphorus and rich soil are beneficial to the growth of fungi or bacteria. However, in the habitat of *T. matsutake*, low pH, soil pH, soil organic matter, nitrogen content and phosphorus content were shown. In addition, the relationship between species is another major factor in the formation and development of microbial communities. Some fungal and bacterial genera could promote the growth of *T. matsutake*, such as the bacterial species of *Burkholderia-Caballeronia-Paraburkholderia* or the fungal species of *Umbelopsis*. We believe that the impact of microbiome composition on the growth and fruiting of *T. matsutake* will be the subject of future studies.

## Figures and Tables

**Figure 1 jof-08-00575-f001:**
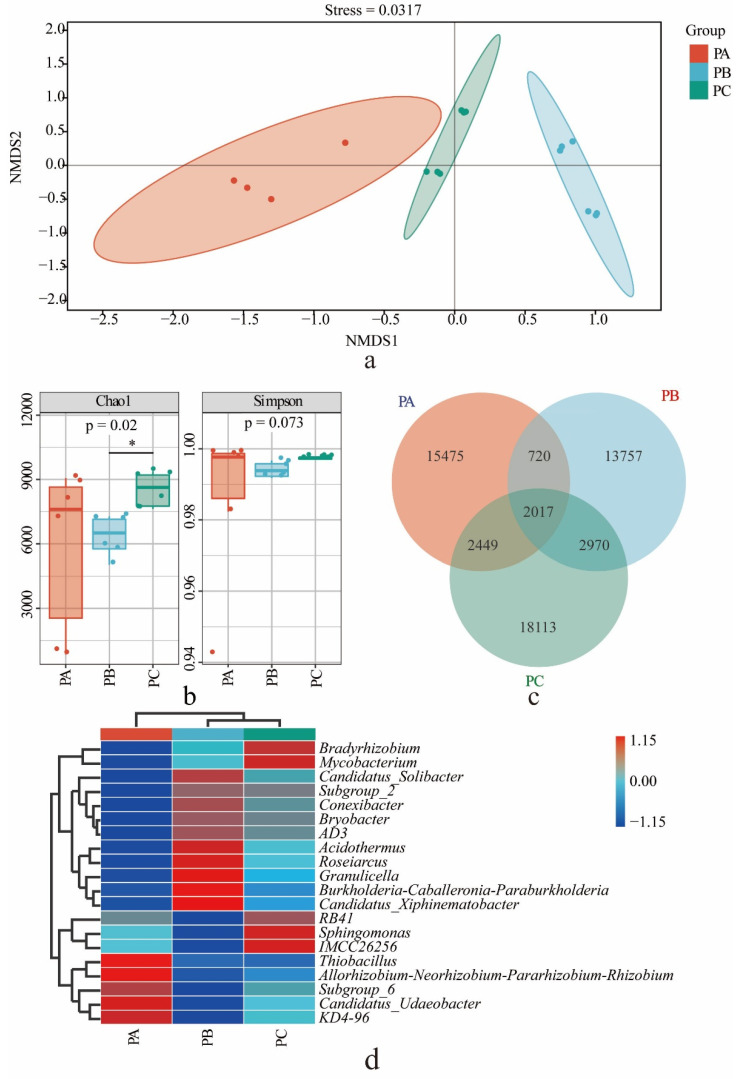
Bacterial community in the soil from three collection sites: (**a**) NMDS analysis of bacterial β-diversity from three collective sites, (**b**) alpha diversity of bacteria, the asterisk (*) represents a significant difference at *p* < 0.05 level, (**c**) Venn map of bacteria, (**d**) a heat map of the bacteria composition of the first 20 genera. PA, soil sample from plot A; PB, soil sample from plot B; PC, soil sample from plot C.

**Figure 2 jof-08-00575-f002:**
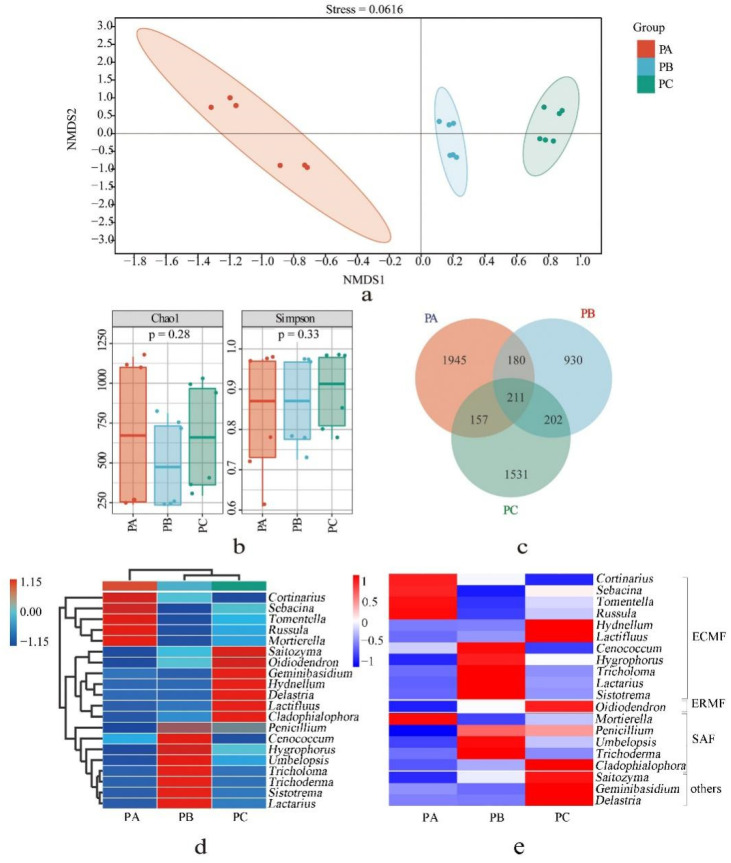
Fungal community in the soil from three collection sites: (**a**) NMDS analysis of fungal beta diversity, (**b**) alpha diversity of fungi, (**c**) Venn map of fungi from three soil samples, (**d**) a heat map of the fungi composition of the first 20 genera. (**e**) Functional guild cluster diagram of fungi of the first 20 genera. ECMF: ectomycorrhizal fungi; ERMF, ericoid mycorrhizal fungi; SAF, saprotrophic fungi. PA, soil sample from plot A; PB, soil sample from plot B; PC, soil sample from plot C.

**Figure 3 jof-08-00575-f003:**
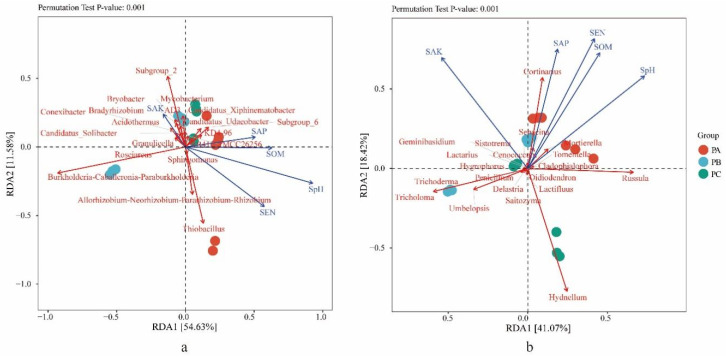
The relationships of soil physicochemical factors with (**a**) bacteria and (**b**) fungal communities in soil with different collection sites. PA, soil samples from plot A; PB, soil samples from plot B; PC, soil samples from plot C.

**Figure 4 jof-08-00575-f004:**
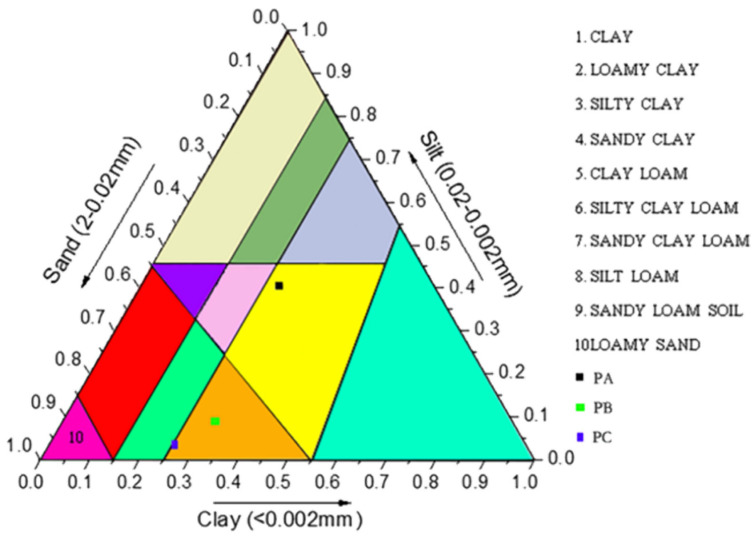
International triangulation of soil texture. PA, soil samples from plot A; PB, soil samples from plot B; PC, soil samples from plot C.

**Figure 5 jof-08-00575-f005:**
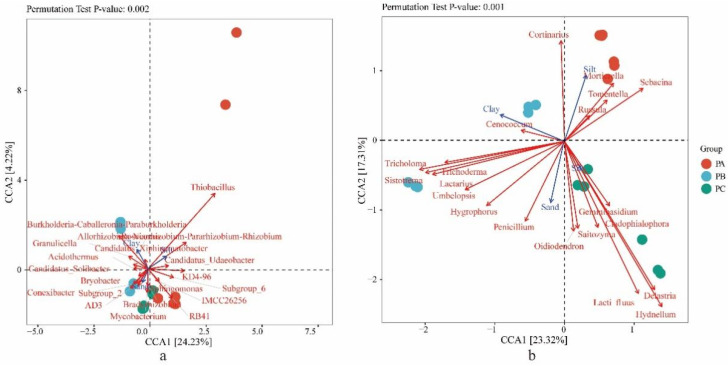
The relationships of soil texture factors with (**a**) bacteria and (**b**) fungal communities in soil with different collection sites. PA, soil samples from plot A; PB, soil samples from plot B; PC, soil samples from plot C.

**Table 1 jof-08-00575-t001:** One-way analysis of variance of abiotic factors in the soil.

	SpH	SOM (mg·kg^−1^)	SAK (mg·kg^−1^)	SEN (mg·kg^−1^)	SAP (mg·kg^−1^)
PA	5.71 ± 0.07 ^a^	106,893.17 ± 9456.2 ^a^	23.57 ± 3.37 ^a^	203.73 ± 9.26 ^a^	20.20 ± 6.34 ^a^
PB	5.00 ± 0.12 ^c^	68,044.84 ± 5118.48 ^b^	25.83 ± 0.58 ^a^	114.32 ± 4.48 ^b^	5.19 ± 1.31 ^b^
PC	5.31 ± 0.06 ^b^	56,300.90 ± 5525.63 ^b^	20.27 ± 1.85 ^a^	94.82 ± 3.06 ^c^	3.87 ± 1.01 ^b^

Note: The data are represented by the mean ± SD, *n* = 3. Different letters in the table represent a significant difference in the alpha diversity indices at the *p* < 0.05 level.

## Data Availability

Sequence data were deposited at the National Center for Biotechnology Information (NCBI) Sequence Read Archive (SRA) database (accession number for the bacterial composition of soil: SAMN17088241–SAMN17088258; and for the fungal composition: SAMN17087468–SAMN17087485). All other data are provided in the Results section and the Appendix A of this article.

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
