# Peer review of "Fungal–Bacterial Networks in the Habitat of SongRong (Tricholoma matsutake) and Driving Factors of Their Distribution Rules"

_jof, 2022, doi:10.3390/jof8060575_

Round 1
Reviewer 1 Report
Abstract
Please explain all abbreviations in a place, where they are introduced for the first time, according to the MDPI instructions for authors. The explanations of abbreviations are also required in the Abstract section (e.g. ITS; rRNA).
Introduction
Lines 34; 36; 40, 45 and so on.
Please, replace the SR code with a scientific name for this species, i.e. "T. matsutake" in the entire manuscript.
Lines 45-46
"The fruiting bodies distribution of SR was closely related to their symbiotic partners, such as Pinaceae and Fagaceae [15,16]".
T. matsutake is widely associated with conifers (Pinaceae), the associations with Fagales are unclear and weakly reported.
Here are more than 400 ITS sequences of T. matsutake in UNITE database:
https://unite.ut.ee/bl_forw_sh.php?sh_name=SH1647740.08FU#fndtn-panel3
Numerous sequences of T. matsutake are recorded as symbionts of Pinaceae species (Pinus, Abies, Picea), including all sequences obtained from ECM roots.
Only two sequences are noted as associated with Fagales (Castanopsis, Quercus), and both are linked to the same paper. I suggest that mentioned dissimilarity could imply methodological errors.
Lines 49-50
Authors wrote, "its fruiting bodies are often found in mixed forests of these species".
That's interesting. A similar pattern was observed for suilloid ECM fungi and conifers. Suilloid species appear frequently in mixed oak-pine species, however, the symbiotic association between Suillus and oaks (i.e. ECM roots, not the occurrence of sporocarps only) was never proved, neither under natural nor artificial conditions.
Interestingly, pine-associated species T. equestre was incorrectly reported as a symbiont of deciduous trees (called Tricholoma equestre var. populinum) due to the morphological similarity. Now it’s known, that T. equestre s.s. is associated with pines only, but poplar-associated sporocarps belong to T. frondosae – similar and closely related, but distinct Tricholoma species. (T. equestre is edible and tasty, T. frondosae is edible, but considered as much less tasty).
Methods & Results
Site description:
"Q. mongolica mixed with Rhododendron dauricum (plot B), and Q. mongolica mixed with R. dauricum and Pinus densiflora (plot C)".
At the same time, the highest share of Tricholoma matsutake was noted on plot B (shown in Table S2 B), but the appropriate tree partner for T. matsutake appeared on plot C. [Rhododendron species do not form ECM associations, they form AM symbiosis only.]
Tricholoma matsutake is repeatedly recorded as an ECM symbiont of conifers (Pinus, Picea, Abies). No conifers are reported for plot B.
Does the site “plot B” contains pine seedlings or mature pines in the surrounding? Even one small conifer tree or large mature conifer tree at a longer distance is sufficient as an appropriate symbiont of pine-associated ECM fungal species.
Please, discuss it carefully.
As long as the ECM roots of T. matsutake are not found on the roots of deciduous trees in the natural/seminatural habitat, and the taxonomic identity of both, T. matsutake and deciduous trees are not proved using DNA analysed of both fungus and the tree, the potential affinity between T. matsutake and Fagales is questionable.
Please, involve in the Introduction and Discussion sections the two mentioned points of view, i.e. putative misidentification of tree hosts for T. matsutake in mixed forests and the potential occurrence of morphologically similar and perhaps closely related Tricholoma species.
Please refer to appropriated papers, if available, with concurrent consideration to the reputation of the journal, including the scientometric indices such as the Impact Factor. Not all journals include peer-review, and some published papers contain crucial methodological errors. Sometimes the sentences are reported by authors as conclusions, even if the authors provide no proof for it, and then it's widely repeated by other authors.
Lines 82-83
"We also analyzed the composition of the species of Tricholoma, but the results are not confirmed due to limitations of the sequencing technology."
Exactly.
Line 94
"We marked five quadrants measuring 20 m × 20 m within a space of more than 100m".
Please, describe all woody plant species inside the quadrants 20x20m (including seedlings) and trees at a distance of 50 meters from the borders of the quadrants, with the special emphasis on the presence of conifers/pines on plot B.
Lines 217-25
“3.2. Fungal community in the soil of SR Tricholoma matsutake habitat”
Please, divide the fungi into ECM clades (e.g. genera Tricholoma, Hygrophorus, Sistotrema, Lactarius, Lactifluus, Russula, Cenococcum, Delastria, Hydnellum, Sebacina, Inocybe, Piloderma, Amphinema, Tomentellopsis) and non-ECM genera. The current classification of ECM fungi here (look at Tab.1 and Fig.1): https://doi.org/10.1007/s00572-009-0274-x
Moreover, the mutual relations between ECM fungi and other, non-ECM fungi (saprotrophs, pathogens) could be interesting (look at Fig. 2 and 3 here: https://doi.org/10.3390/f12030353 )
Figure 1, point (d); Figure 2, point (d)
Please, explain why on a heat map of the composition of the first 20 fungal genera the negative values are presented. It is not possible for the fungal composition. Each taxon can be absent (0) or present (positive values). Is this map a corrplot? If yes, it needs to be clarified in the description.
Lines 307-310
“Nine fungal genera, such as Russula (r = -0.832, p = 0.001), Inocybe (r = -0.710, p = 307 0.001), Leotia (r = -0.781, p = 0.000), Entoloma (r = -0.666, p = 0.003), Trechispora (r = -0.695, p 308 = 0.001), Cylindrosympodium (r = -0.638, p = 0.004), Ilyonectria (r = -0.695, p = 0.001), and 309 Clitocybe (r = -0.617, p = 0.006) were negatively correlated with Tricoloma”
Please, replace Tricoloma with Tricholoma.
The first mentioned genera, i.e. Russula, Inocybe, Leotia and Entoloma, contain ECM fungi. However, it’s hard to say, whether the negative correlation reflects the competition between ECM fungi or rather the environmental dissimilarity between plots/sites.
DISCUSSION
Please, replace SAP (soil available phosphorus) with “phosphorus content”.
Please, replace SOM (soil organic matter) with “soil organic matter”.
Please, replace SEN (soil effective nitrogen) with “nitrogen content”.
Please, replace SAK (soil available potassium) with “potassium content”.
Please, replace SpH (soil pH) with “soil pH”.
Please, replace the above-mentioned abbreviations in the entire manuscript, except figures*. It makes the text more readable.
* - Due to the limited number of letters available for the titles of variables (e.g. in R software), the abbreviations can be necessary.
Line 386
“This pattern is very similar in all groups of organisms in other environmental conditions”
Please, refer to the other studies on ECM fungi on the roots of trees and the functional redundancy of ECM assemblages.
Line 386
“R. dauricum may be promoted the establishment of a symbiotic relationship between Q. mongolica and SR T. matsutake.”
First, in this study, the authors provided no proof for the putative symbiotic affinity between Quercus and T. matsutake.
Second, Rhododendron species are associated with AM, not ECM fungi, thus their <promotion of the establishment of symbiosis between Quercus and T. matsutake> is highly unlikely.
The presence of R. dauricum can either, impact the soil and then, indirectly, the environmental conditions for ECM fungi (including T. matsutake), or can be a result of environmental dissimilarity. Otherways, suitable habitats for T. matsutake can overlap with suitable habitats for R. dauricum, and it does not imply the impact of one species on another.
The co-occurrence of non-ECM plant species and ECM fungal species under the same environmental conditions cannot be treated as equivalent to the occurrence of interspecific relations between these species.
Even the co-occurrence of ECM tree species and ECM fungal species under the same environmental conditions cannot be treated as equal to the occurrence of symbiotic interactions between these species.
For example, Suillus species commonly occur in mixed oak-pine forests. Suillus species are widespread symbionts of pines, but no symbiotic affinity between Suillus and oaks was ever proved.
References
Please, involve the relevant studies conducted on other continents than Asia, like Europe and North America. Numerous papers on the ectomycorrhizal assemblages between ECM fungi and trees based on the analyses of ECM root tips and/or meta sequencing of soil samples was published in the last five years.
Please refer to the relevant papers, paying attention to the reputation of the journal, including the scientometric indices such as the Impact Factor.
Author Response
Dear editor and reviewer:
We are especially grateful for your patience and generous comments on the manuscript, and we have edited the manuscript to address your concerns. The details are in the attachment. We believe that the manuscript is now suitable for publication in the Journal of Fungi. Please see the attachment.
Thanks again.
Kind regards,
Ruiqing Ji
On behalf of all authors.

Reviewer 2 Report
I reviewed the manuscript „Fungal-bacterial networks in the habitat of SongRong (Tricholoma matsutake) and driving factors for their distribution rules“. I found it is an interesting research and inovative articl.
General comment: can be accepted after major revision.
English should be much better. I suggest that the article be reviewed by a native speaker.
Introduction can be much better!
some examples:
line 43-45 - you can't write in the first person!
line 50-52 - you can't write in the first person!
line 82-83 - this is unnecessary if you cannot confirm the results!
The method are well chosen, statistics is OK!
Results:
The sentences in the results are poorly and clumsily written, ie they are not easy to read with a lot of grammatical errors. I suggest correcting the sentences in the results!
Conclusions
The presentation of the figure in the conclusion is unnecessary and does not contribute to the quality of the article!
Almost 50% of references are older than 10 years!
Author Response

(The authors gave the same response as above.)

Reviewer 3 Report
In this work, the authors sought useful information about the relationships between fungal and bacterial networks and various influence factors in the SongRong Tricholoma matsutake habitat. The research was set up, studying 3 sites with different natures of soils, Mycorrhizal plant partners and environmental factors that can affect the growth and development of SR, using as a control sampling site the pure forest of Quercus mongolica without SR. The experimentation was well performed and the methods adopted as High-throughput sequencing and statistical methods, are reliable and effective, to provide new insight on the Fungal-bacterial networks in the different habitats of SR.
The screening of bacterial and fungal community in the soil of SR habitat is useful and interesting.
The most important results indicate that soil fungi-bacterial networks in the habitat soil of SR (Plot B and Plot C), including species abundance, diversity, particularly in terms of β-diversity and species composition are different from the control site Plot A.
Although the analysis of soil microbiological data is complex, what emerges is the greater influence of physico-chemical characteristics and texture properties that influenced the abundance of fungi and bacteria in the soil rather than the host itself.
There is little to object to the correctness of the work carried out. I invite the authors to provide some clarification.
- Can the authors indicate more specifically the physical and chemical characteristics that most affect the abundance and diversity of soil microflora?
- Have the authors identified an optimal pH range and climatic conditions of the soil for the expansion of its microflora?
- Do the authors know if the plant species present in the three sites emit radical exudates that can promote or limit the expansion of the soil microflora?
- From the experiments conducted, can the authors better clarify whether the nutritional status of the soil as a source of macro and micro nutrients can affect the microbial expansion and the symbiotic state between fungus-bacterium and root? and why?
- The authors performed the cellulase test on microorganisms positively related to Tricholoma?
- the authors at the conclusion of their work, could they suggest which forest site and habitat is most suitable for the growth and development of SR, Tricholoma matsutake?
Author Response
Dear editor and reviewer:
We special thank again for your patience and generous comments on the manuscript, and we have edited the manuscript to address their concerns. The details are in the attachment. We believe that the manuscript is now suitable for publication in the Journal of Fungi. Please see the attachment.
Thanks again.
Kind regards,
Ruiqing Ji
On behalf of all authors.

Round 2
Reviewer 2 Report
I reviewed the manuscript “Fungal-bacterial networks in the habitat of SongRong (Tricholoma matsutake) and driving factors for their distribution rules”. I found it is an interesting research and inovative article.
General comment: can be accepted in this form.
All my comments are accepted in the new version of the article!